
# Ground motions variability in Israel from 3-D simulations of
# M 6 and M 7 earthquakes
Jonatan Glehman[1] and Michael Tsesarsky[1,2]
[1]Department of Earth and Environmental Sciences, Ben Gurion University of the Negev, 8410501, Israel
[2]Department of Civil and Environmental Engineering, Ben Gurion University of the Negev, 8410501, Israel
*Correspondence to*: Jonatan Glehman (glechman@post.bgu.ac.il)
**Abstract.** In Israel, due to low seismicity rates and sparse seismic network, the temporal and spatial coverage of
ground motion data is insufficient to estimate the variability of moderate-strong (M > 6) ground motions required
to construct a local ground motion model (GMM). To fill this data gap and to study the ground motions variability
of M > 6 events, we performed a series of 3-D numerical simulations of M 6 and M 7 earthquakes. Based on the
results of the simulations, we developed a statistical attenuation model (AM) and studied the residuals between
simulated and AM PGVs and the single station variability. We also compared the simulated ground motions with
a global GMM in terms of peak ground velocity (PGV) and significant duration (Ds 595). Our results suggest that
the AM was unable to fully capture the simulated ground motions variability, mainly due to the incorporation of
super-shear rupture and effects of local sedimentary structures. We also show that an imported GMM considerably
deviates from simulated ground motions. This work sets the basis for future development of a comprehensive
GMM for Israel, accounting for local sources, path, and site effects.
## 1    Introduction
The recent report by the Centre for Research on the Epidemiology of Disasters (CRED) and the UN Office for
Disaster Risk Reduction (UNDRR) – Human Cost of Disasters, 2000 - 2019 – clearly shows that earthquakes are
the deadliest natural disasters. Counting only 3 % of the total number of people affected by natural disasters, they
count for 58 % of deaths (more than 700,000) of all disaster types and 21 % of recorded economic losses (Mizutori
& D'ebarati, 2020). Over the past 40 years, the global population exposed to a moderate to severe intensity
earthquake has increased by 93 % (to 2.7 billion people) (Pesaresi et al., 2017). This value is expected to grow
with population growth and increasing urbanization.

Seismic hazard is the intrinsic natural occurrence of earthquakes and the resulting ground motion and other
effects (Wang, 2005). Ground motion models (GMM's) are critical components in the mitigation of seismic
hazard. Empirically based GMMs, also known as Ground Motion Prediction Equations (GMPE's), are parametric
models that estimate the median and the variability of the expected ground motions at a site. The main explanatory
variables of such models are typically earthquake magnitude, distance, and site conditions. New generation
GMPEs also address faulting style, depth to rock, and others.

Many regions worldwide, either due to low seismicity rates and/or sparse coverage of the seismic network,
do not provide sufficient temporal and spatial data to estimate the variability of ground motions required to
construct a local GMPE or validate an imported GMPE to local conditions. This situation is specifically acute in
the range of strong earthquakes at relatively short distances that pose the most significant hazard to human life
and infrastructure.



The use of imported GMPE's under the ergodic assumption attributes the ground motion variability to the
randomness of the process (i.e., aleatory variability) rather than to local systematic source-path and site effects
(i.e., epistemic uncertainty) (Anderson & Brune, 1999). Abrahamson et al., (2019) showed that the increased
number of strong-motion records over the past decade exhibit significant differences in scaling of the ground
motions even within relatively small regions and that most of the variability typically treated as aleatory is actually
due to systematic source, path, and site effects. Kuehn et al., (2019) showed the importance of variations in quality
factor (Q) over small spatial scales (30 km) in California. Specifically showing that accounting for path effects
leads to a smaller value of the aleatory variability and results in different median predictions, depending on source
and site location. To achieve this improvement, Kuehn et al., (2019) divided California into a grid with a cell size
of 30 km by 30 km and used 12,039 records from 274 events recorded at 1504 stations. This approach can be
employed only in data-rich regions, such as California. Lan et al., (2019) showed that for South Western China,
imported GMPEs result in significant discrepancies compared with regional instrumental data (including the
Wenchuan Mw 7.9 event). In addition, despite the recorded ground motion data expanding, it remains sparse for
large, complex ruptures with recurrence intervals generally exceeding the observation length of instrumental
records.
The challenges met while predicting ground motion in data-poor regions turn numerical modeling into an
essential complementary method for seismic hazard analysis (Chaljub et al., 2010). Numerical modeling alleviates
the need for the ergodic assumption, as it can augment the seismic data with strong motion records and account
for ground motions variability by systematically separating source, path, and site effects. For example, Graves et
al., (2011) showed that the combination of rupture directivity and basin response effects could lead to an increased
hazard in particular sites, relative to that calculated by GMPE. Pitarka et al., (2021) found that the combination of
rupture propagation effects with the amplification due to local topography can result in large ground motions
amplifications with complex spatial variability.
However, the shift from ergodic models to nonergodic models, which account for local source-site and
path effects such as numerical models, leads to large epistemic uncertainty in the median ground motion, resulting
in increased epistemic uncertainty of the hazard (Walling & Abrahamson, 2012). Such uncertainty derives from
both modeling and parametric uncertainties, as the model is not well constrained. However, model uncertainty
can be reduced by using more accurate 3D crustal models and source models.
Subsurface models with different levels of accuracy and completeness, are available for other parts of the
world. With the increasing use of terrestrial and space geodesy, the control of seismic sources is also improving
with time. Combining the two enables the construction of numerical models for regional assessment of ground
motions (Douglas & Aochi, 2008; Graves & Pitarka, 2015; Pitarka et al., 2021). A hybrid GMM, based on
empirical and synthetic ground motion databases, is expected to reduce the epistemic uncertainty of the median
ground motion and will lead to a lower aleatory variability than magnitude limited GMPE's.
In Israel, low seismicity rates (centennial and millennial return periods) and a limited instrumental catalog,
span only four decades and contain mainly M < 6 events, impede the development of local empirical GMM. The
practical outcome of this shortcoming is the use of imported GMM's, such as the Campbell & Bozorgnia, (2008)
used in the Israel Seismic Design Code IS 413 (Israel Standards Institution, 2013). Contrary to the instrumental
catalog, the Israel pre-instrumental catalog spans over three millennia (Agnon, 2014), including numerous M > 6





events, with up to 14 M > 7 events. In addition, recent geodetic studies (Hamiel et al., 2016; Sadeh et al., 2012)
identified a slip deficit on specific segments of the Dead Sea Transform (DST) equivalent to an M > 7 earthquake.
This paper presents numerical modeling of ground motions in Israel, intended to narrow the strong ground
motion data gap and study ground motions variability from moderate (M 6) and strong (M 7) earthquakes. We
begin with a brief introduction to the seismo-tectonic setting of the region. Then, we proceed to the methodology
section to describe the process of generating a synthetic ground motion database and the subsequent construction
of a statistical ground motion model. The results section presents the simulated ground motions and the respective
attenuation model. Then, it compares it with the global GMPE's of  Campbell & Bozorgnia, (2014; hereafter,
CB14) and Afshari & Stewart, (2016) performance with respect to the synthetic database. Finally, we discuss our
findings and provide insights regarding the seismic hazard from moderate to strong earthquakes and the
importance of developing a comprehensive regional GMM to mitigate the seismic hazard in Israel.

## 2    The seismo-tectonic setting of Israel

### 2.1    Seismicity and seismic hazard in Israel

The Dead Sea Transform (DST) fault system is an active tectonic boundary separating the African and Arabian
plates. Extending from the Gulf of Aqaba to southern Turkey, a total length of ca. 1100 km, it dominates the
seismicity of Israel, Palestinian Authority, Lebanon, and Syria (Fig. 1a,b). The DST is a left-lateral strike-slip
fault with a total offset of 105 km (Garfunkel, 2014). The average long-term slip rate is  4 to 5 mm year$^{-1}$ (Bartov
et al., 1980). Geodetic slip rates along the Israel part of the DST ranges from 3 to 5 mm year$^{-1}$ (Hamiel et al.,
2016; Sadeh et al., 2012).
Splaying north-west from the DST is the Gilboa Fault, and farther north-west towards the Mediterranean,
the Carmel Fault. Both comprise an active zone generalized as the Carmel Fault Zone (CFZ). The DST segments
are capable of producing M 6 and M 7 events (Shamir et al., 2001), and the CFZ is capable of producing M > 6.5
earthquakes (Grünthal et al., 2009).
The Israel Seismic Network (ISN), established in 1983 and upgraded over the years, consists of a mixture
of different instrumental and operational stations, including short-period stations (24 in total), broadband stations
(14 in total), and a large broadband array (part of the Comprehensive Nuclear Test Ban Treaty). The deployment
of the ISN does not cover areas of increased seismic hazard, e.g., densely populated zones and soil sites, or areas
designated by the Israel Seismic Code (IS413) as suspected in extreme ground motion amplification, such as the
Zevulun Valley (Fig. 1b). Currently, the seismic network is upgraded within the Tru'a project (an early warning
system), with up to 60 strong-motion accelerometers and 12 broadband seismometers added to ISN (Kurzon et
al., 2020). However, most of the instrumentation will be placed along the DST and Carmel fault to provide early
warning, and not in densely populated or industrialized areas where the seismic risk is tangible. Based on
demographic projections (the Taub Center for Social Policy Study in Israel) the population of Israel is expected
to grow from 9.05 million in 2021 to 12.8 million in 2040 and combined with the increasing demand for housing
and infrastructures, the seismic risk is expected to grow.
The Israel seismic catalog covers 36 years of measurements (1985–2021) and includes more than 23,300
events (Wetzler & Kurzon,2016), but only 15 of them are of M > 5 (Fig. 1a and Fig. 2). Moving back in time,
Israel's pre-instrumental catalog spans over 3000 years (Agnon, 2014; Zohar, 2019) with many catastrophic


events, such as the 749 (M > 7), 1202, (M > 7.5), 1759 (M > 7), and the 1837 (M >7) earthquakes, among others.
In total, fourteen M > 7 events were cataloged by Ambraseys (2006) in the past two millennia. Recent geodetic
studies (Hamiel et al., 2016; Sadeh et al., 2012) identified a slip deficit on specific segments of the DST, such as
the Jordan Gorge Fault (JGF) and the Jordan Valley Fault (JVF), equivalent to an M > 7 earthquake.

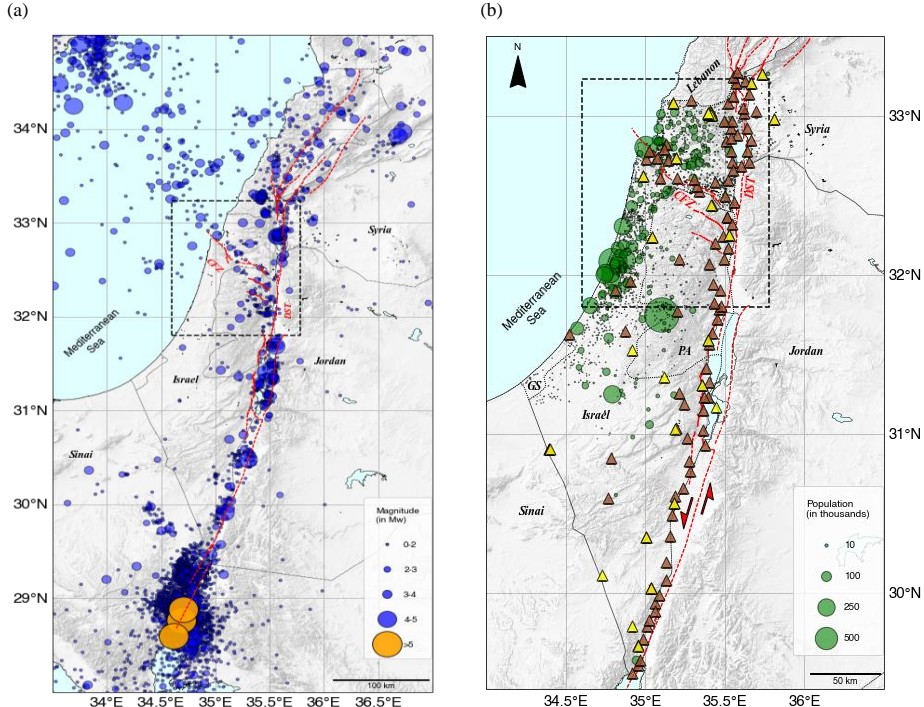

**Figure 1.** (a) Israel Seismic catalog (Mw) for the period 1985-2021 orange circles are events with Mw > 5 (expansion of
Wetzler & Kurzon (2016) catalog). Red lines are active tectonic borders and faults, DST is Dead Sea Transform, CFZ is
Carmel Fault Zone. (b) Demographics of Israel and the Palestinian Authority and the deployment of the Israel Seismic
Network. Yellow triangles are the old (up to October 2017) Israel Seismic network, brown triangles are the current (Tru'a)
seismic network. (after Kurzon et al., (2020)). GS is Gaza Strip. The black rectangles define the computational domain
presented in Fig. 3a.

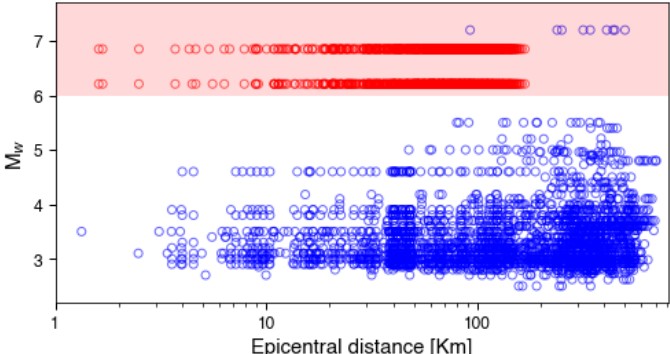


**Figure 2.** Israel's ground motion database (blue circles) for the period 1983-2021 as a function of epicentral distance (Yagoda-
Biran et al., 2021). The shaded rectangle spans the Mw > 6 region of moderate-strong ground motion records. The red circles
are the simulated ground motions from this work.

### 2.2 Spatial heterogeneity of Israel

The geological structure of Israel exhibits strong spatial heterogeneity over short scales (Fig. 3a,b). Deep pull-
apart basins (up to 10 km) filled with soft sediments ($Vs \sim 600\text{-}800$ m sec$^{-1}$) accompany the active DST system,
from south to north: The Dead Sea Basin, Beit Shean Valley (BSV), the Sea of Galilee (SG) and the Hula Valley.
Along the CFZ, the Zevulun, Harod, and Jezreel Valleys are formed. The vulnerability of Zevulun Valley is
particularly crucial because of its dense population and the high concentration of strategic industrial infrastructure
(Shani-Kadmiel et al., 2020).

The Israeli coastal plain is one of the most densely populated regions of the country (on average, 9000
people per km$^2$), is underlain by a westward thickening sedimentary wedge (SW). In the Judea foothills area, east
of the SW, a strong reflector exists between the sandstones and clays (Pleistocene Kurkar Gr, $Vs \sim 300$ m sec$^{-1}$)
and the hard carbonate rocks (the Cretaceous Judea Gr., $Vs \sim 2000$ m sec$^{-1}$). In the coastal plain, the Kurkar Gr.
overlays the soft carbonates (Avedat Gr, $Vs \sim 900$ m sec$^{-1}$) and clastic sediments (the Bet Guvrin Fm., $Vs \sim 800$
m sec$^{-1}$) (refer to Fig. 3b). The depth of the Kurkar Gr. base reflector is typically several tens of meters. Further
to the west, a prominent reflector is a contact between the clays (Pliocene Yafo Fm., $Vs \sim 600$ m sec$^{-1}$) and top of
Judea Gr. These two reflectors, when shallower than 250 m, were used for the latest update of the Israel Building
Code IS 413 (Israel Standards Institution, 2013) to delimit areas of high potential of ground motion amplification
(Gvitzman & Zaslavsky, 2009). This situation further complicates the process of developing an empirical new
generation GMM for Israel.

### 2.3 Source effects

The impact of inter-basin sources along the DST on regional ground motions was examined by Shimony et al.,
(2021). This work clearly showed that regional ground motions are determined by source-path coupling effects in
the strike-slip basins before waves propagate into the surrounding areas. Ground motions are determined by the
location of the rupture nucleation, the near-rupture lithology, and the local structures. Shimony et al., focused on
symmetric sub-shear ruptures and did not model rupture directivity or super-shear rupture velocities, both known
to amplify regional ground motions.

Under specific conditions, super-shear ruptures and directivity occur on bi-material faults (Shi & Ben-Zion
2006). Specifically, for subsonic propagation, symmetrically initiated bilateral rupture evolves after some
propagation distance to a unilateral rupture in the positive direction, which is the direction of slip on the compliant
side of the fault containing the softer layer. The magnitude of this effect increases with propagation velocity and
the degree of material contrast across the fault. At super-shear propagation speeds, along a bi-material fault, the
propagation direction is reversed.
The DST is a mature left-lateral fault with a 105 km offset, resulting in strong material contrast between
the hard layers on the Jordan side (east) and the soft layers on the Israeli side (west). Thus, the rapture can
potentially propagate unilaterally southwards, discharging most of the seismic energy into Israel or in super-shear
mode. The Jordan Gorge Fault and the Jordan Valley Fault (both active faults of the DST) specifically can produce
an earthquake with rapture propagating in super-shear velocity since they border deep sedimentary basins,
characterized by large shear wave velocities contrast along the rapture propagation path. Thus, to quantify the
seismic hazard ensuing from bi-material faults, it is necessary to study the two propagation directions and supe-
shear velocities.
The primary purpose of this study is to examine different source-path and site effects of simulated,
moderate M 6 and moderate-strong M 7 earthquakes and their contribution to ground motion variability. To this
end, we simulated M 6 and M 7 earthquakes with different source and path properties. Then, we developed a
statistical representation of median ground motions and their variability. Formulated in terms of Peak Ground
Velocity (PGV), our model quantifies the spatial distribution of the ground motions in central and northern Israel,
accounting for local source, path, and site effects, including rupture velocity and directivity.

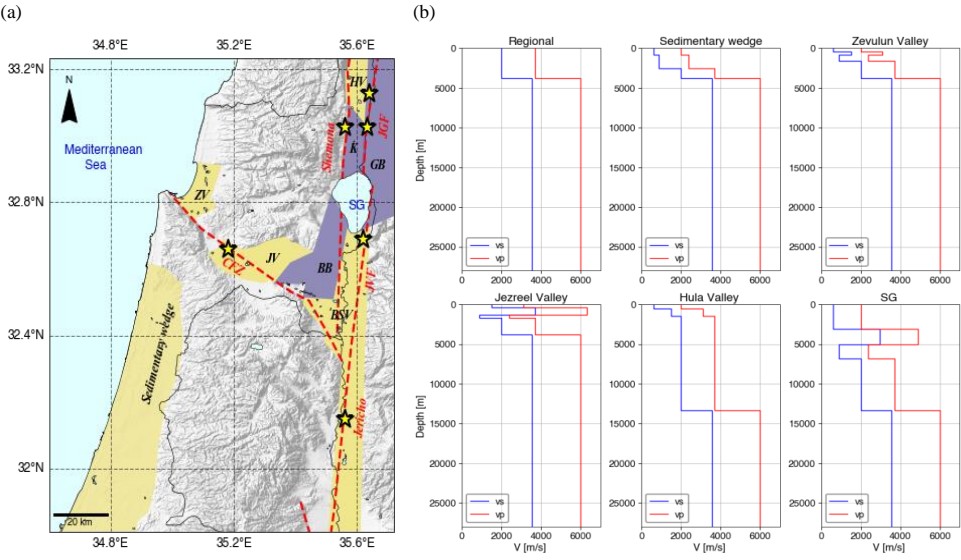

**Figure 3.** (a) The DST fault system and the Carmel Fault Zone (CFZ) and accompanying structures. BSV-Beit Shean Valley,
ZV-Zevulun Valley, JV-Jezreel Valley, HV-Hula Valley, SG-Sea of Galilee, K-Korazim structural saddle, BB-Belvoir Basalts,
GB-Golan Basalts, and the Sedimentary wedge. The yellow stars indicate the epicenter of the seismic sources simulated in our
work: Jordan Gorge Fault (JGF), with bilateral and unilateral slip realization, Jordan Valley Fault (JVF), Jericho Fault,
Shemona Fault (only for M 7), and CFZ (only for M 6). (b) Representative depth velocity profiles of the computational domain.



## 3     Methodology and workflow
Developing a regional GMM for Israel requires a database of ground motions records, including M > 6 events at
short, <100 km, distances. To supplement the existing ground motions database, we added a suite of synthetic
ground motions from physics-based 3D numerical models of different M 6 and M 7 earthquakes (Fig. 2).
Our work comprised two main stages; first, we developed the regional velocity model of Shimony et al.,
(2021), following we simulated five different earthquake scenarios for each magnitude, with nucleation at
different locations along the DST and CFZ. For each scenario, we recorded synthetic ground motions at 129
stations (see supplementary material, Fig. S1). Next, we performed a statistical analysis on the synthetic database
by minimizing residuals between data and model estimations. We then formulated a statistical model of the ground
motions and examined its consistency with the simulated database.
### 3.1     Numerical model
Ground motions in this research were modeled using the SW4v2 software (Petersson & Sjogreen, 2014, 2017a,
2017b), developed for large-scale simulations of seismic wave propagation on parallel computers.
The velocity model covers the northern and central part of Israel (fig. 4a) and includes the main DST trough
and the following basins/structures, from south to north: Beit Shean Valley (BSV), Belvoir Basalts (BB), Sea of
Galilee (SG), Korazim structural saddle (K), Golan Basalts (GB) and Hula Valley (HV). Along the CFZ, we model
the major sedimentary basins of Jezreel Valley (JV) and Zevulun Valley (ZV). The coastal plain is underlain by
the westward thickening Sedimentary wedge (SW). Geographically, the model extends from the city of Ashdod
in the south (31.8° N, 34.6° E) to Hula Valley in the north (33.23° N, 35.72° E) and from the Mediterranean Sea
in the west to the Golan Basalts in the east. Figures 4b,c,d illustrate the north-south and east-west cross-sections
of the velocity profiles. The numerical domain spans 159 km in the north-south direction and 124 km in the east-
west direction. It covers almost 80 % of the Israeli population and a significant part of the population of the
Palestinian Authority.
Subsurface geometry and the characteristics of the DST trough were obtained from Rosenthal et al., (2019)
with modifications for the Hula Valley, obtained from the density log of the Notera 3 (Rybakov et al., 2003). The
sedimentary wedge structure retrieved from Gvirtzman et al., (2008) and The Zevulun Valley structure was set
using data from Gvirtzman et al. (2011). The basement depth along the model is based on Ben-Avraham et al.,
(2002). . Five physical quantities describe the viscoelastic material model used in this research: shear wave
velocity (Vs), pressure wave velocity (Vp), density ($\rho$), and seismic quality factors (Qs, Qp) for each point in the
computational space. The missing parameters were assessed indirectly by using the correlation presented by
Brocher (2008). The main units with their respective velocity, density and quality factors are shown in Table 1.
Seismic sources were modeled using the distributed slip model (DSM) developed by Shani-Kadmiel et al.,
(2016). DSM is a kinematic model which describes the rupture patch as an elliptic surface with maximum slip at
the nucleation point, decaying toward the edges as a pseudo-Gaussian function (Fig. S2). Rupture patch size and
displacements were scaled following the relations presented in Wells & Coppersmith (1994). All sources were
modeled as left-lateral, vertical strike slips (a dip of $90°$ and rake of $0°$), with a strike of $3°$ for sources on the
DST and a strike of $325°$ for the CFZ. The moment-rate time function of each point on the rupture patch was set
to a GaussianInt pulse (Petersson & Sjogreen, 2017b) with a central frequency of $f_0$=0.4 Hz and a maximum
frequency of $f_{max}$=1 Hz.





The depth of the model was set to 28 km corresponding to the maximum seismogenic depth in this region
(Wetzler & Kurzon, 2016). We assigned a minimum shear wave velocity of 608 m s$^{-1}$ for the uppermost
sedimentary layer due to the computational limitations of our system. Grid spacing was set to 76 m in accordance
with the minimum shear wave velocity and the maximum frequency of the source. We set the simulation time to
120 seconds to allow the slowest waves to propagate across the entire computational domain. The main parameters
of the numerical setting are summarized in Table 2.





**Figure 4.** (a) The numerical model of the computational domain accompanied with subsurface cross-sections, marked with red dashed lines: (b) east-west-cross section through Zevulun Valley, CC' (c) east-west cross-section through the Sedimentary wedge, BB' and (d) north-south cross-section through the DST trough, AA'.


**Table 1.** Material properties of main stratigraphic units used in this work

| Model part | Rock Formation | Vs [m s⁻¹] | Vp [m s⁻¹] | Qs | Qp | ρ [Kg m⁻³] |
|---|---|---|---|---|---|---|
| Regional | Crystalline basement | 3550 | 6000 | 403 | 806 | 2720 |
| | Cenozoic and Senonian sediments (Judea/ Talme Yafe, Mount Scopus Avedat, and Lower Saqiye) | 2000 | 3700 | 160 | 320 | 2350 |
| Local variations: | | | | | | |
| DST | Cenozoic sediments (Umm Sabune, Bira and Gesher) | 887 | 2380 | 62 | 124 | 2054 |
| | Miocene volcanics (lower basalt) | 3698 | 6330 | 439.5 | 879 | 2790 |
| | Pliocene volcanics (upper basalt) | 2947 | 4900 | 282 | 564 | 2520 |
| | Notera/Lisan | 608 | 2000 | 39.87 | 79.74 | 1900 |
| Hula | Cenozoic sediments | 1500 | 3100 | 111.5 | 223 | 2245 |
| | Notera/Lisan | 608 | 2000 | 39.87 | 79.74 | 1900 |
| JV | Cenozoic sediments (Umm Sabune, Bira, and Gesher) | 887 | 2380 | 62 | 124 | 2054 |
| | Miocene volcanics (lower basalt) | 3698 | 6330 | 439.5 | 879 | 2790 |
| | Cenozoic sediments | 1500 | 3100 | 111.5 | 223 | 2245 |
| ZV | Cenozoic and Senonian sediments (Mount Scopus Avedat and Beit Guvrin) | 887 | 2380 | 62 | 124 | 2054 |
| | Cenozoic sediments (Patish) | 1500 | 3100 | 111.5 | 223 | 2245 |
| | Cenozoic sediments (Kurkar and Yafo) | 608 | 2000 | 39.87 | 79.74 | 1900 |
| SW | Cenozoic sediments (Lower Saqiye) | 887 | 2380 | 62 | 124 | 2054 |
| | Cenozoic sediments (Kurkar and Upper Saqiye) | 608 | 2000 | 39.87 | 79.74 | 1900 |

**Table 2.** Main parameters of the numerical model

| Parameters | Value |
|---|---|
| Model Dimensions (L×W×D) | 159.63 Km × 124.45 Km × 28 Km |
| Spatial spacing (dh) | 76 m |
| Grid size (points) | $1.27 \times 10^9$ |
| Time step spacing | 0.0125 s |
| Simulated time | 120 s |
| Source Dimensions (L×D) | M 6: 16 Km × 8.5 Km |
| | M 7: 19 Km × 8.5 Km |
| Source maximum and average slip | M 6: 0.5 and 0.2 m |
| | M 7: 3 and 1.3 m |
| Seismic moment ($M_0$) | M 6: $2.57 \times 10^{18}$ N·m (Mw 6.21) |
| | M 7: $2.37 \times 10^{19}$ N·m (Mw 6.85) |
| Source fundamental ($f_0$) and maximal frequencies ($f_{max}$) | 0.4 and 1 Hz |

**3.2    Earthquake scenarios and database**
To examine the variability of ground motions from moderate M 6 and strong M 7 earthquakes, we concentrated
on earthquake events nucleating on active segments of the DST system, with known slip deficit, and along the


CFZ. We modeled a symmetric bilateral rupture on the Jordan Gorge Fault (JGF-B), Jericho Fault (JF) Carmel
Fault Zone (CFZ) and the Shemona Fault (SF), a southward unilateral rupture on the JGF (JGF-U), and a super-
shear rupture on the Jordan Valley Fault (JVF) (Fig. 3).
The hypocenter for the DST events was placed in the middle of the seismogenic depth; 11 and 13 Km, for
the M 6 and M 7 respectively, for the M 6 CFZ, the value was set to 12 Km. The rupture patch was designed to
be contained in uniform lithology to prevent super-shear rupture speeds in the shallow parts of our model.
Therefore, rupture speed for each scenario was set to 0.9 $V_S$ of the lithology surrounding the nucleation zone. The
only exception was the JVF scenario for both M 6 and M 7, in which we modeled super-shear effects. The rupture
velocity of each scenario corresponds to the local variations of the sediment's depth. Following the transition of
the nucleation zone from the shallow crystalline basement in the south and west parts of the model to the thick
Mesozoic and Cenozoic sediments in the north and the east, the rupture velocity decreases from 3195 m s$^{-1}$ along
the Shemona, Carmel, and Jericho faults to 1800 m s$^{-1}$ along the JGF and JVF faults. As a reference, we simulated
a simple two-layered reference model (Ref) on the JGF, with mechanical properties similar to the regional setting,
following Aldersons et al., (2003). The scenarios are summarized in Table 3.
**Table 3.** Earthquake scenarios

| Fault Name | Scenario | Magnitude (M) | Rupture speed (m s$^{-1}$) | Hypocentral depth (Km) |
|---|---|---|---|---|
| Jordan Gorge | Bilateral rupture (JGF-B) | 6, 7 | 1800 | 11 and 13 |
| Jordan Gorge | Southward unilateral rupture (JGF-U) | 6, 7 | 1800 | 11 and 13 |
| Jordan Valley | Bilateral super-shear rupture (JVF) | 6, 7 | 1800 | 11 and 13 |
| Jericho | Bilateral rupture (JF) | 6, 7 | 3195 | 11 and 13 |
| Shemona | Bilateral rupture (SF) | 7 | 3195 | 13 |
| Carmel | Bilateral rupture (CFZ) | 6 | 3195 | 12 |
| Reference | Bilateral rupture (Ref) | 6, 7 | 3195 | 11 and 13 |

**4    Results**
In this section, we report the simulation results and the simulation-based attenuation model for M 6 and M 7. We
begin with elaborating on the regression process and its deliverable, the attenuation model. Next, we show the
correspondence of the model with the simulated database in terms of PGV residuals and examine the contribution
of each earthquake scenario to the total deviation. Then, we proceed with looking into single station variability,
through maps of the predicted and simulated PGV, with the corresponding residuals at each station. Finally, we
examine the PGV and the 5 %- 95 % ground motions significant duration (Ds 595) correspondence between
predicted by global GMM's (CB14, Afshari & Stewart, 2016, respectively) and simulated.
**4.1    Simulation results**
For each simulation, we attained a set of 129 synthetic ground motion records (3 components each) from the
network deployed in the computational domain. Next, we calculated the PGV values for each scenario at each
station. We decided to exclude some of the M 7 near-source records (stations: 104,105 and 106 for the JVF





scenario and stations: 122,123 and 129 for the JGF-B, JGF-U, and Shemona scenarios) due to high strain values
and possible non-linear effects, not compatible with the linearity assumption of our model. In total, our ground
motions database consists of 645 and 633 synthetic records for M 6 and M 7, respectively. Figure 5 presents our
results in terms of PGV as a function of distance. We use different markers for records from the sedimentary
structures of the Zevulun Valley and the Sedimentary wedge to differentiate them from the remaining data.

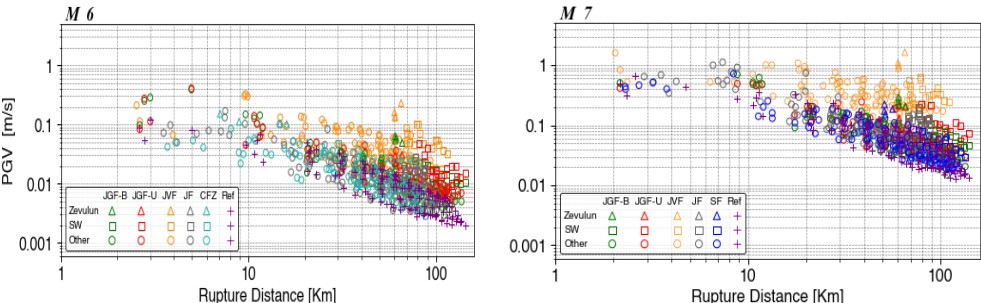

**Figure 5.** Simulation results, PGV-distance space, for M 6 (left) and M 7 (right). The records from Zevulun Valley and the
Sedimentary wedge (SW) are marked with triangles and rectangles, respectively. The other records are marked with circles;
the reference records are marked with pluses.

### 4.2    Statistical analysis of ground motions results

The next step was to formulate a statistical ground motion attenuation model (AM) for the two magnitudes based
on our simulations. Such a model will provide an estimate for the median ground motions and their variability.
The general parametric form of the AM for both M 6 and M 7 is presented in Eq. (1):
$$\ln Y = a \ln\left(\sqrt{R_{RUP}^2 + b}\right) + c \ln\left(\frac{V_{s,surf}}{V_{s,min}}\right) + d\,Z_2{}^2 + e \; \pm \sigma \tag{1}$$
Where Y is ground motion parameter. Due to the bandwidth of our numerical models (0.1 to 1 Hz), we formulated
the AM in terms of PGV. We use the closest distance to the fault rupture plane ($R_{RUP}$ as defined in CB14) as the
initial explanatory variable. To improve the accuracy of the model, we incorporated two additional variables into
the regressions: surface shear wave velocity at the site ($V_{S,\,surf}$) and the depth to $V_S = 2$ km s$^{-1}$ ($Z_2$), which is the
depth to the hard Mesozoic sediments (top Judea Gr.) considered the primary reflector in the region. *a, b, c, d,* and
*e* are model coefficients, and σ is the standard deviation. The $V_{S,\,min}$ is the minimum shear wave velocity in the
computational domain, which in our model equals 608 m s$^{-1}$.
The process of minimizing the residuals as a function of each explanatory variable can be found in the
supplementary material (Fig. S3). We used $V_{S,\,surf}$ instead of the more common $V_{S30}$, as our grid resolution is 76
m, preventing us from accurately determining the time-averaged shear wave velocity in the top 30 m of each site
in our model. The coefficients and the total standard deviation for each model are summarized in Table 4.
**Table 4.** Regression coefficients for the attenuation model (AM)

| Magnitude | IM | a | b | c | d | e | Standard Deviation ( σ) |
|---|---|---|---|---|---|---|---|
| 6 | PGV | -1.01 | 59.34 | -0.685 | 0 | 0.56 | 0.727 |
| 7 | PGV | -1.4 | 257.43 | -0.782 | -0.02 | 4.08 | 0.737 |



**4.3 AM Variability**
Following, we examined the simulated data and the contribution of each scenario to the AM variability. We
calculated the within-event ($\delta W$) and between-event ($\delta B$) residuals (see Al Atik et al., (2010)) for each magnitude
and distance:
$\delta W_{i,j} = \ln PGV_{i,j}^{sim} - \ln PGV_i^m$           (2)
$\delta B_i = \ln PGV_i^m - \ln PGV^{AM}$           (3)
where $PGV_{i,j}^{sim}$ is the simulation value for event i and recording j, $PGV_i^m$ is the median for event i, and $PGV^{AM}$ is
the AM median value. The total residual is the sum of the within and between event residuals.
The residuals are presented in Fig. 6: total (Fig. 6a and 6b), within-event (Fig 6c and 6d), and between-
events (Fig, 6e, and 6f). The within-event residuals (Fig. 6c,d) do not exhibit apparent bias or trend. However,
some of the between-event residuals (Fig. 6e,f) exhibit distance dependency. Most clearly, for M 7, the JVF
(super-shear model) and JGF-U (directivity model) residuals increase with rupture distances greater than 30 km.
The JVF residuals also demonstrate the same distance dependency for M 6; however, the effect is less prominent
when compared to M 7. The JGF-U does not exhibit an apparent trend for M 6. The total residuals (Fig. 6a and
6b) show a large underprediction of the PGV from the JVF scenario (orange) on which we modeled a super-shear
rupture, up to a ratio of almost 2.5 in the Zevulun Valley (orange triangles), for both magnitudes. However, the
AM also exhibits over predictions; The PGV from the scenarios nucleated in the crystalline basement (SF, JF, and
CFZ), with rupture speed= 3195 m s$^{-1}$), are overpredicted down to a ratio of almost -1.5.
For the JGF scenarios, as shown in Fig 5, there is a tradeoff between the ground motion intensity in the
Zevulun Valley (triangles) and the Sedimentary wedge (rectangles). While in a symmetric rupture (JGF-B), the
seismic energy dissipates equally into the north and south parts of the model in an asymmetric rupture (JGF-U),
more energy propagates toward the south, resulting in stronger ground motions at the Sedimentary wedge.
However, the ground motions are less intensive at the Zevulun Valley compared to the symmetric rupture. As a
result, the within-event residuals for Zevulun Valley are higher for the JGF-B scenario compared to the JGF-U
scenario, while for the Sedimentary wedge, the opposite is true.

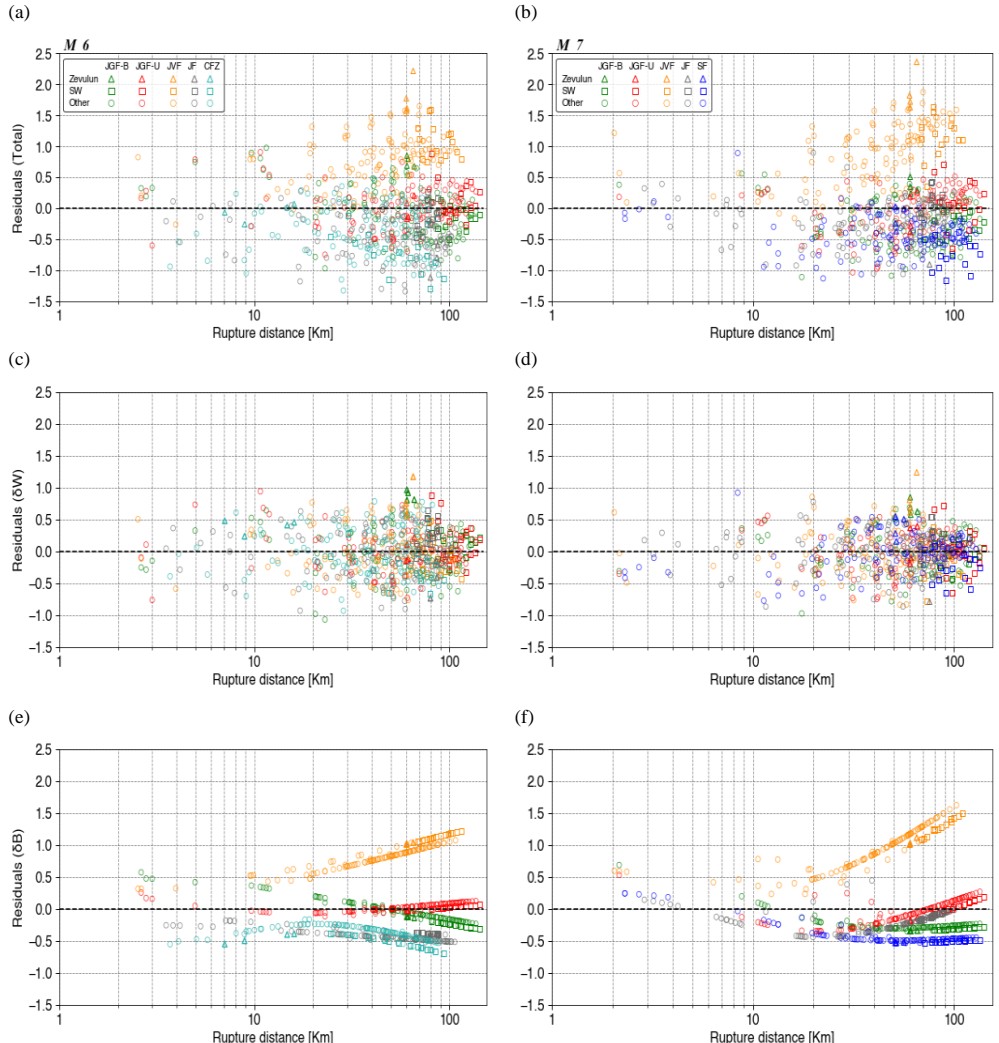

**Figure 6.** Residuals between simulated and attenuation model (AM) PGV as a function of rupture distance ($R_{RUP}$), for M 6 (left) and M 7 (right); (a) and (b) total residuals, (c) and (d) within-event ($\delta W$) residuals, (e) and (f) between-event ($\delta B$) residuals. The records from Zevulun Valley and the Sedimentary wedge (SW) are marked with triangles and rectangles, respectively. The other records are marked with circles. Residuals are in ln units.

We further study the single station variation of ground motions and quantify the misfit between the simulated PGV and the AM PGV. We calculate the mean ground motion and its standard deviation at each station. The residuals for single station k were calculated as follows:

$$\delta_k = \ln PGV_k^{\ sim} - \ln PGV_k^{\ AM} \tag{4}$$

where $PGV_k^{\ sim}$ and $PGV_k^{\ AM}$ are the simulated and predicted mean PGV at station k, respectively. Figure 7 and Figure 8 show the mean simulated and mean AM PGVs for M 6 and M 7, respectively. For each station, we also plot the standard deviation using a scaled diameter circle.





Both figures show that simulated ground motions variability at a single station is large, not fully covered
by the AM. For example, simulated ground motions at station 129 (for location, please refer to Fig S1) exhibit a
significant standard deviation. For M 6, it is the largest value (green triangle) of 0.17 m s$^{-1}$ compared to 0.09 m s$^{-1}$
(indigo) predicted by the AM, while for M 7, the largest standard deviation is 0.59 m s$^{-1}$ (orange triangle)
compared to 0.02 m s$^{-1}$ (light green triangle) observed at station 127 located on the Zevulun Valley (for location,
please refer to Fig S1). As a result, there is a large discrepancy between the simulated and AM values.
In general, higher mean PGV values are accompanied by a larger standard deviation for both magnitudes;
however, the ground motions variability is larger for M 7.

### 4.4      Comparison with global models

To examine the agreement between our simulations with an instrumental, global GMM, we calculated the total
residuals between PGVs from our simulations and PGVs predicted by the CB 14 model. Figure 9 shows the total
residuals for the AM and CB14 models as a function of distance ($R_{RUP}$). For both magnitudes, the AM (mean and
standard deviation) oscillates near the zero-model bias (black horizontal dotted line). However, it deviates when
approaching the region containing rupture distances typical of the Zevulun Valley. The effect is more noticeable
for M 7. Figure 9 also shows that the CB14 is less consistent and performs differently for each magnitude. While
for M 6, the GMPE mostly over predicts (negative values) the simulated PGV (until reaching ZV and SW rupture
distances zones), for M 7, it mostly under predicts them (positive values), except for large distances, up to a factor
of 2 and above. In addition, the CB-14 exhibits a significant standard deviation of the mean ground motion, with
considerably larger variability for M 7.
It is important to note that, by averaging the PGVs, we subdue the performance of both models at individual
stations/Rupture distances; thus, we cannot analyze the residual's spatial variations at a specific location.
However, it is sufficient to demonstrate that the global model deviates considerably from simulated ground
motions.

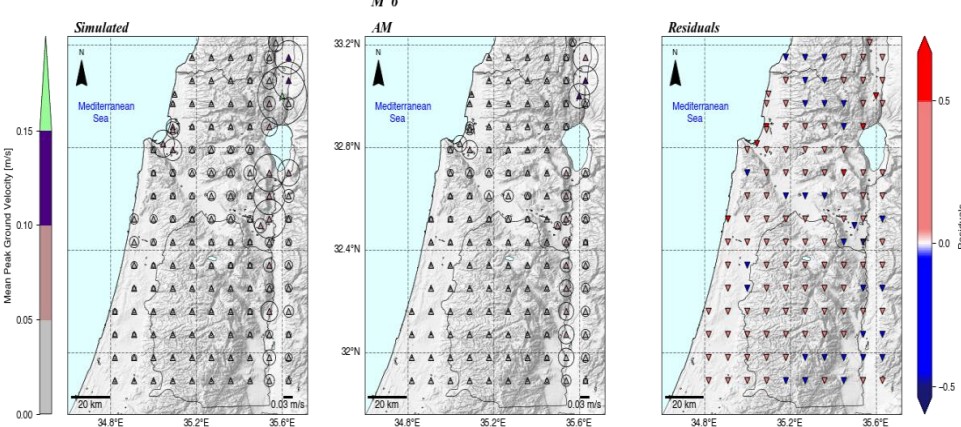

**Figure 7.** Map view of simulated and AM mean PGV (triangles) for M 6 and their standard deviation (diameter of the circles)
at each station, with the respective residuals in ln units (inverted triangles).


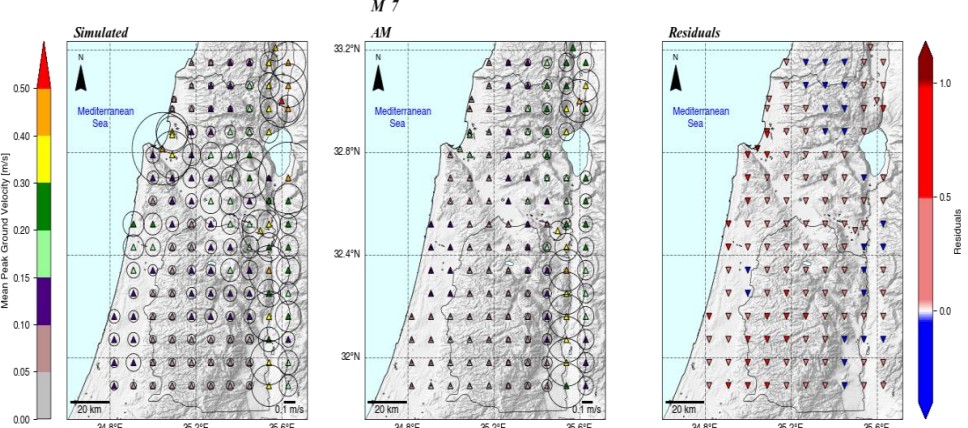

**Figure 8.** Map view of simulated and AM mean PGV (triangles) for M 7 and their standard deviation (diameters of the circles) at each station, with the respective residuals in ln units (inverted triangles).

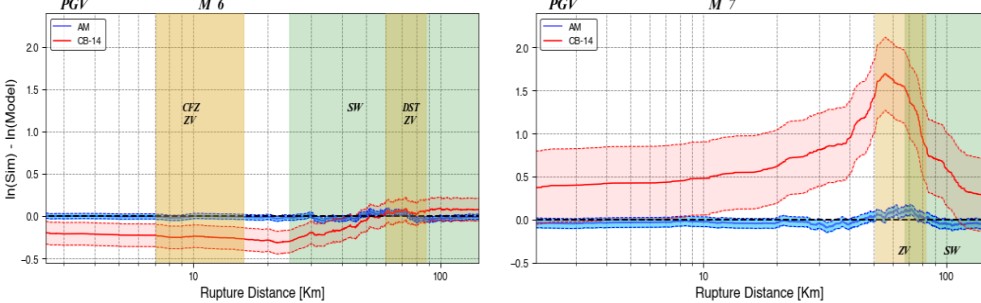

**Figure 9.** PGV Residuals between simulated (Sim) and predicted by the AM (blue) and CB-14 (red) models, as a function of rupture distance ($R_{RUP}$), for M 6 (left) and M 7 (right). Thick lines represent the mean, and the shaded region denotes the standard deviation at each distance. The green and yellow shaded regions indicate the range of rupture distances related to the Sedimentary wedge (SW) and the Zevulun Valley (ZV), respectively. Residuals are in ln units.

### 4.5 Significant duration

Another important intensity measure is the significant duration (Ds595), the time interval between 5 % to 95 % of the cumulative seismic energy (Arias Intensity) at a site. Figure 10 shows the simulated and empirical Ds 595 values as a function of rupture distance. The typical increase of the empirical model with distance is captured in the reference (laterally homogenous) model. However, for all other models, the significant duration remains nearly constant, at ruptures distances larger than 20 km. In addition, the empirical GMM mostly under-predicts the simulated values between 2 to 50 Km for both magnitudes.

We postulate that this is caused by the complex geological setting of our model. The impact of geological complexity is reflected in Ds 595 values from Zevulun Valley (triangles) and the Sedimentary wedge (rectangles). The energy accumulates faster in these structures than in other sites, as the ground motions are amplified, reaching 95 % of the total energy over a shorter duration. Interestingly, the significant duration in Zevulun Valley is lower than in the Sedimentary wedge. As we expect from deep sedimentary structures to prolong shaking duration, it



may sound counterintuitive. However, it is explained by the relative proximity of the Zevulun Valley to the
rupture. Whereas in Zevulun Valley, most of the energy arrives as a pulse at the beginning of the record, the
energy at the more distant Sedimentary wedge accumulates more gradually and reaches its maximum almost at
the end of the record, resulting in higher Ds595 values. In general, there is no large deviation between the
simulated significant duration for M 6 and M 7. However, the empirical model shows a longer duration for M 7.

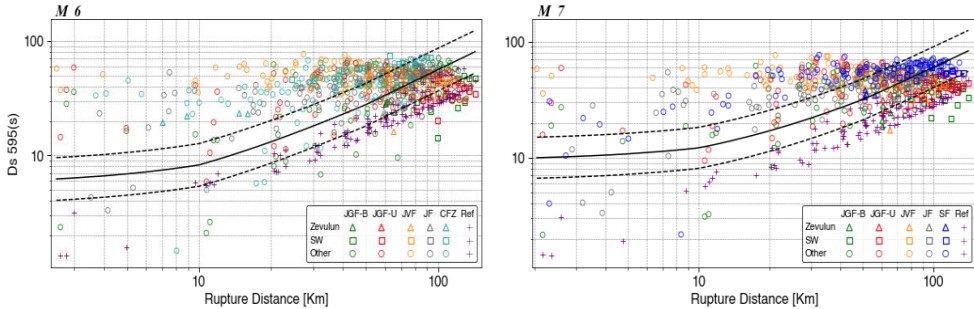

**Figure 10.** Comparison of 5 % to 95 % ground motions significant duration (Ds 595) between simulated and empirical GMM
(Afshari & Stewart, 2016), for M 6 (left) and M 7 (right). Solid and dashed lines represent the median and the standard
deviation of the empirical GMM, respectively. The records from Zevulun Valley and the Sedimentary wedge (SW) are marked
with triangles and rectangles, respectively. The other records are marked with circles.

## 5   Discussion

A strong earthquake in Israel is imminent. However, up to date, a comprehensive regional GMM describing the
spatial variability of ground motions has not yet been developed. This is mainly due to low seismicity rates and
magnitude bounded strong motion database, coupled with sparse instrumental coverage. The current ground
motion database lacks events with magnitude M > 6. To fill this gap and examine different source and path effects
on ground motions variability, we simulated M 6 and M 7 earthquakes with different source and path properties.
Subsequently, to study the ground motions variability, we developed a statistical attenuation model (AM) of PGV
for M 6 and M 7 earthquakes, based on $R_{RUP}$, $Z_2$, and $V_{S, surf}$ explanatory values.
Our analysis shows that the AM was unable to fully capture the variability of the simulated ground motions.
Except for the Jordan Valley Fault (JVF) scenarios, the AM overestimates most of the modeled ground motions.
We postulate that this overestimation results from the outlier, higher PGV values from the JVF scenario (Fig. 5),
shifting the average ground motion toward them. Also, the residuals for the JVF scenario show a distance
dependency for $R_{RUP}$ > 20 Km, continuing to grow away from the fault. We describe this scenario as a "black
swan" of our simulations and account its outlier behavior to the effects of the super-shear rupture, specific to this
model. Super-shear ruptures behave differently from sub-shear ruptures in many aspects. Most pertinent to our
analysis is the slow energy decay of the super-shears relative to sub-shears (Bhat et al., 2007); thus, it cannot be
fully captured by our AM, which is based mainly on sub-shear ruptures. In addition, it was found that $Z_2$, depth to
Mesozoic rock, is not a good predictor for the M 6. As a result, the M 6 model depends only on rupture distance
and $V_{S, surf}$.



For each scenario, both magnitudes considered, we observed high PGV values at the Zevulun Valley and
the Sedimentary wedge associated with local site effects. These sedimentary structures exhibit a larger
discrepancy between the simulated and AM PGV values when compared with other sites. Such deviation indicates
that the AM does not fully capture the site effects of these complex structures, and future model refinements are
required. Likewise, the single station variability shows that the simulated values' highest mean and standard
deviation were in Zevulun Valley and near-source stations. In addition, a relatively high standard deviation was
also found in the Sedimentary wedge for M 7. This large single station variability is, apparently, the impact of the
outlier JVF PGV values. The AM does not account for the standard deviation at near-source and Zevulun Valley
stations for the M 6 and almost at all stations for the M 7. In fact, as the AM was unable to capture the simulated
JVF PGV values, it is expected that the single station variability cannot be captured either. Furthermore, we show
that the larger discrepancy for M 7 is due to the larger deviation of the JVFs ground motions from the mean (Fig.
6d,e).
Noteworthy to mention that while the effect of the super-shear rupture on the AM performance is
systematic over the entire computational domain, the effect of the southward directivity is distance-dependent,
increasing towards the south, related to a larger amount of energy discharged in this direction. Additional records
of super-shear and directivity ruptures will improve the performance of the AM and will assist in better
understanding the implications of these phenomena on the seismic hazard in Israel.
The comparison of the simulated ground motions with a global GMM model (CB-14) showed that this
model is not well constrained for the simulated ground motions and does not capture their total variability. We
note that the comparison was performed on a single IM, the PGV values, one of several intensity measures
provided by the CB-14. Thus, our findings are pertinent to the variability of PGV solely. It should be noted that
PGV is a good proxy for structural damage (e.g., to Kaestli & Fäh, (2006); Wald et al., (1999)), hence a crucial
parameter for seismic hazard mitigation. This discrepancy between modeled PGV and CB-14 PGVs, will
inevitably result in a discrepancy in the evaluation of structural damage.
The significant duration (DS595) comparison showed again that the imported model performs differently
than the simulated ground motion and cannot explain the local variability due to complex geological structure,
affecting the source-path and site terms of the ground motions.
We acknowledge that our AM is not independent of the evaluated models, thus describing both their
explanatory and predictive power (Mak et al., 2017). However, our goal was not to develop an independent and
comprehensive GMM but to study the ground motion variability through a statistical ground motion model.
Recently, Maiti et al., (2021) developed a suite of nine GMMs for Israel, in the magnitude range of 3 to 8
and distance range of 1 to 300 Km. These models are formulated in Fourier amplitude spectra (FAS) and are based
on one empirical and four simulated ground motions datasets and two empirical host models. The simulated
ground motions were generated using the Stochastic Method SIMulation (SMSIM) model of Boore (2003), with
a unique set of parameters for each simulation, calibrated with the empirical ground motions dataset (discussed in
detail in Yagoda-Biran et al., (2021)). However, the GMMs do not fully account for a local source, path, and site
effects due to sparse empirical database at large magnitudes (M > 6) and the utilization of a point-source stochastic
simulation method. This method is useful for simulating mean ground motions. Yet, it is less appropriate for
simulating site-specific and earthquake-specific ground motions and low-frequency ground motions, which are
affected by the 3D geometry of the computational domain. The AM presented in this work is based on 3D



simulations and incorporates a finite fault source with different rupture properties. This is the first step toward
developing a regional GMM, accounting for local source, path, and site effects. In subsequent work, which is
beyond the scope of the current research, we intend to develop a complete GMM for Israel, which will include all
the magnitudes and will be based on empirical (M < 6) as well as on synthetic (M > 6) databases. Such a model
is expected to perform better than imported global models by maintaining both; a lower aleatory variability and,
as new synthetic data will be added to the database, reduced epistemic uncertainty of the median ground motions
(Abrahamson et al., 2019).
To summarize, the population of Israel is fast-growing, with an annual rate of 1.8 % (OECD 2020 data),
compared with the 0.4 % average of the OECD. Coupled with fast economic growth of 4.5 % (OECD 2019 data),
the demand for housing and infrastructure constantly elevates the seismic risk in Israel. Our work shows that the
ground motions in Israel from M 6 and M 7 earthquakes are expected to be very damaging, up to 8-9 EMS (Fig.
S4). Furthermore, the modeled ground motions exhibit considerable spatial variability, which imported GMMs
do not fully capture. The development of a local comprehensive GMM model is therefore critical for the mitigation
of seismic risk. In the foreseen future, the moderate-strong ground motion data gap will be filled by synthetic
ground motion records from systematic numerical simulations.
**Data and resources**
Israel Seismic catalog (Fig. 1a), expanded after Wetzler & Kurzon (2016) catalog and the configuration of the
Israel    seismic    network    (Fig.    1b)    after    Kurzon    et    al.,    (2020)    can    be    found    at
https://earthquake.co.il/en/earthquake/searchEQS.php and https://earthquake.co.il/en/network/
accNetwork.php, respectively. The ground motions database of Israel (Fig. 2) discussed in Yagoda-Biran et al.,
(2021) is available at https://earthquake.co.il/en/hazards/EngSeismology.php. The Taub Center population
projections for Israel are accessible at https://www.taubcenter.org.il/en/pr/population-projections-for-israel-2017-
2040/. OECD population and economic growth rates can be found at https://data.oecd.org/israel.htm#profile-
economy. Simulations were performed using SW4 version 2.0 (v2.0; Petersson and Sjögreen, 2017a), an open-
source package for wave propagation simulations, available at github.com/geodynamics/sw4 (last accessed June
2021). Data processing was done with the pySW4 package from Shahar Shani-Kadmiel, available at
https://github.com/shaharkadmiel/pySW4 (last accessed July 2021), and "obspy" (Beyreuther et al., 2010),
developed for numerical seismology. Figures were prepared with Matplotlib (Hunter, 2007) and Cartopy (Met
Office, 2016). Peak ground velocity (PGV) values, according to Campbell and Bozorgnia (2014), were calculated
using the Next Generation Attenuation-West Project (NGA-West2) ground-motion prediction equations (GMPEs)
excel file, available at https://apps.peer.berkeley.edu/ngawest2/databases/ (last accessed July 2021). The
supplemental material includes: (1) synthetic station network deployed in our models (Fig. S1); (2) distributed
slip model (DSM) slip distribution and rupture time (Fig. S2); (3) the evolution of the residuals between simulated
and attenuation model (AM) PGV for M 6 and M 7 (Fig. S3) and (4) map view of simulated mean EMS intensity
calculated according to Kaestli & Fäh, (2006).
*Competing interests*. The authors declare that they have no conflict of interest.



**Acknowledgments**

This research was partially funded by the Ministry of Energy, Israel (Grant Number 219-17-02). Co-author JG was partially supported by the Ministry of Energy scholarship for graduate studies (Tender 76/19).

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
