# Peer review of "Ground motions variability in Israel from 3-D simulations of M 6 and M 7 earthquakes"

_Natural Hazards and Earth System Sciences, 2021_

## Referee Comment (RC1)

Referee comments for manuscript nhess-2021-280 titled "Ground motions variability in Israel from 3-D simulations of M6 and M7 earthquakes", by Jonatan Glehman and Michael Tsesarsky

**General comments**

The manuscript describes a study of ground motion variability associated with simulations of 3D ground motions from M7 and M6 earthquakes, along major seismogenic sources in Israel. The authors use a 3-D model of the sub-surface that describes the main features of the spatial heterogeneity in Israel, and simulate the ground motions at 129 measurement locations in the northern part of Israel. Then the authors derive a statistical attenuation model based on the simulations, and analyse the residuals, single station variability and significant durations.

This is of importance to low-moderate seismicity regions, and specifically to Israel. Complementing the instrumental catalogue with simulated strong ground motions is important, as well as understating and evaluating the variability associated with the simulated ground motions.

**Specific comments**

Why is there no subfigure for the minimization of the residuals to all 3 predictor variables in Figure S3 for magnitude 6?

When comparing the simulations to CB14 – can the authors elaborate as to the predictor variable values used in CB14? Did they use 608 m/s for Vs30?

It seems like the AM over-predicts GM at the sedimentary wedge rupture distances (Figure 9), can the authors comment on that? It would seem that most of the data at these distances is lower, and will result in a lower model that under-predicts.

Lines 354 – 363: I really don't see this in the data. I don't see the triangles lower than the squares, and it is very difficult to tell them at all from the other symbols. I suggest thinking of a clearer way to present the data, as presently the figure does not support the statements in the text.

In lines 389 – 391, the authors state that the AM wasn't able to capture the full site effects of the Zevulun Valley and the sedimentary wedge, and further model refinements are required. Do the authors think that such effects can be incorporated successfully into a regional GMM?

**Technical corrections**

The acronyms GMM and GMPE are used throughout the manuscript, please select one and be consistent.

The past and present tense are used interchangeably to describe what is  / was done. Please select one and be consistent.

Line 70 – what is a magnitude limited GMPE? Unclear. Please explain.

Line 93: Israel should be Israeli.

Line 93: ranges should be range.

Figure 1 caption (lines 121, 123): I think it should say Israel seismic network **stations**.

Line 135: "the Israeli coastal plain**,** "  - the comma is missing, and the word "is" should be deleted.

Line 143: delineate rather than delimit.

Lines 159 – 166: the word rupture is misspelled many times.

Line 179: "ground motion**s** records"- the 's' should be deleted.

Line 182: the sentence "we developed the regional velocity model of Shimony et al (2021) is unclear. If the model is Shimony et al.'s – then it wasn't developed by the authors. If it is a modification, or based on their results, then please explain what was done.

Line 183: the word "following" here seems awkward. Please consider rephrasing.

Line 185 – "statistical analysis **of** the synthetic database".

Line 203: in the Gvirtzman et al. quote the comma should be deleted. Also, the capital T in The Zevulun Valley is unnecessary.

Line 255: station 123 is not visible in Figure S1.

Line 281 – it seems like there should be a better way to start a sentence than "Following, ". Perhaps: "We then examine the simulated…".

Page 14 – Fig. 6 – it seems that the SF scenario is missing from the figure altogether, but referred to in the text. This is also true for figure 5. I also recommend that in the captions of figures 5 and 6 the abbreviations of the faults and source characteristics be explained again.

Line 299: the tradeoff sentence is a bit unclear (what trades off here). I suggest adding: "…. between the ground motion intensity in the Zevulun Valley (triangles) and the *ground motion intensity* in the Sedimentary wedge….".

Line 300 – a comma is missing before "…in an asymmetric rupture…"

Figure 7 – the color bar of the mean peak ground velocity isn't very visible, I advise you to pick different colors, and make them vivid, say green – yellow – orange- red. Right now it is very difficult to observe the colors. Also, since you give just two locations as examples (station 129 and stations 127), it would be more convenient to identify these in the figure, rather than refer to a figure in the supplementary material.

Lines 369-375: this paragraph seems more suited to a summary section.

Line 399: the word is is missing: "noteworthy to mention **is** that…."

Line 404: when using the GMM acronym, the word model is redundant.

Line 406: define IM.

Line 419: "four simulated ground motion**s** datasets" – the s of the plural for ground motion should be omitted.

---

## Author Comment (AC1)

Re: Authors Reply to Review nhess-2021-280

Dear reviewer,

We would like to thank you for reviewing our paper, *"Ground motions variability in Israel from 3-D simulations of M 6 and M 7 earthquakes"* (*nhess-2021-280*).

Thank you for the time spent and the positive attitude towards this work. Your remarks and critique certainly improved the manuscript.

Attached below is our pointwise reply to the general and specific comments. All the changes and additions are marked in the revised version of the manuscript.

All the co-authors approved the revision. Please address all correspondence to:

*Jonatan Glehman*

Department of Earth and Environmental Sciences
Ben Gurion University of the Negev
Beer-Sheva 8410501
Israel.

(glechman@post.bgu.ac.il)

Referee comments for manuscript nhess-2021-280 titled "Ground motions variability in Israel from 3-D simulations of M6 and M7 earthquakes", by Jonatan Glehman and Michael Tsesarsky

**General comments**

The manuscript describes a study of ground motion variability associated with simulations of 3D ground motions from M7 and M6 earthquakes, along major seismogenic sources in Israel. The authors use a 3-D model of the sub-surface that describes the main features of the spatial heterogeneity in Israel and simulate the ground motions at 129 measurement locations in the northern part of Israel. Then the authors derive a statistical attenuation model based on the simulations, and analyse the residuals, single station variability and significant durations.

This is of importance to low-moderate seismicity regions, and specifically to Israel. Complementing the instrumental catalogue with simulated strong ground motions is important, as well as understating and evaluating the variability associated with the simulated ground motions.

**Specific comments**

Why is there no subfigure for the minimization of the residuals to all 3 predictor variables in Figure S3 for magnitude 6?

**Reply:** For M6, $Z_2$ has a minimal impact on the residual's standard deviation (<0.001), therefore we decided to not include it as an explanatory variable (please refer to table 4 and lines 406-408 on page 18); Hence, there was no subfigure for $Z_2$ in figure S3 for M 6.

However, to clarify this point further, we followed your suggestion and added the quantitative impact of $Z_2$ on M 6 and the impact of Z2 on the minimization of the total residuals in Figure S3 (a,c and e). Thank you. Please refer to lines 404 - 407 in the revised manuscript.

When comparing the simulations to CB14 – can the authors elaborate as to the predictor variable values used in CB14? Did they use 608 m/s for Vs30?

**Reply:** Thank you for this comment. The CB-14 PGVs were calculated for a strike-slip fault, where we used the surface shear wave velocity as the Vs30 parameter input for CB-14 (this is due the minimum grid spacing of 76 meters). The Basin response term $Z_2$ in our model is analogue to $Z_{2.5}$ in CB-14. The Magnitude term was calculated according to the simulated magnitudes; Mw 6.21 and Mw 6.85. We added this information and an example of CB-14 GMM to Figure 5 to clarify this point further, please refer to lines 270 – 271 (Figure 5. Caption) and 337-339 in the revised manuscript.

It seems like the AM over-predicts GM at the sedimentary wedge rupture distances (Figure 9), can the authors comment on that? It would seem that most of the data at these distances is lower, and will result in a lower model that under-predicts.

> **Reply:** Thank you for this valuable comment. The residuals between the simulated and AM (mean PGVs in Figures 7 and 8 show that, except for stations 18 and 17 (located on the edge of the Sedimentary wedge), the AM under-predicts the ground motions at the Sedimentary wedge. Thus, the contribution to the AM's over-prediction comes from other sites located at these distances. As mentioned on page 15, lines: 348-351, the residuals were computed as a moving average over distance. Thus, they are independent of site-specific terms.

Lines 354 – 363: I really don't see this in the data. I don't see the triangles lower than the squares, and it is very difficult to tell them at all from the other symbols. I suggest thinking of a clearer way to present the data, as presently the figure does not support the statements in the text.

> **Reply:** Thank you for pointing it out. We modified figure 10 to support the text in the lines mentioned above.

In lines 389 – 391, the authors state that the AM couldn't capture the full site effects of the Zevulun Valley and the sedimentary wedge, and further model refinements are required. Do the authors think that such effects can be incorporated successfully into a regional GMM?

> **Reply:** This is a challenging question. We do believe that such effects could be incorporated into a regional GMM. We are currently studying additional terms; unique to the Zevulun Valley and the Sedimentary wedge, such as $Z_{0.8}$ as well as distance-dependent and rupture velocity-dependent attenuation terms, for directivity and super-shear ruptures, among others. We added this information to the revised manuscript. Please refer to lines 462 - 464.

**Technical corrections**

The acronyms GMM and GMPE are used throughout the manuscript, please select one and be consistent.

> **Reply:** Thank you for this comment. We changed all acronyms to GMM.

The past and present tense are used interchangeably to describe what is / was done. Please select one and be consistent.

> **Reply:** Corrected. Thank you.

Line 70 – what is a magnitude limited GMPE? Unclear. Please explain.

> **Reply:** Explained. Thank you.

Line 93: Israel should be Israeli.

> **Reply:** Corrected. Thank you.

Line 93: ranges should be range.

> **Reply:** Corrected. Thank you.

Figure 1 caption (lines 121, 123): I think it should say Israel seismic network **stations**.

> **Reply:** Corrected. Thank you.

Line 135: "the Israeli coastal plain**,** " - the comma is missing, and the word "is" should be deleted.

> **Reply:** Corrected. Thank you.

Line 143: delineate rather than delimit.

> **Reply:** Corrected. Thank you.

Lines 159 – 166: the word rupture is misspelled many times.

> **Reply:** Corrected. Thank you.

Line 179: "ground motion**s** records"- the 's' should be deleted.

> **Reply:** Corrected, line 181. Thank you.

Line 182: the sentence "we developed the regional velocity model of Shimony et al (2021) is unclear. If the model is Shimony et al.'s – then it wasn't developed by the authors. If it is a modification, or based on their results, then please explain what was done.

> **Reply:** Thank you for highlighting this point. The model of Shimony et al., was refined and expanded. We added this information to the text, lines 184-186 in the revised manuscript.

Line 183: the word "following" here seems awkward. Please consider rephrasing.

> **Reply:** Corrected, line 186. Thank you.

Line 185 – "statistical analysis **of** the synthetic database".

> **Reply:** Corrected, line 188. Thank you.

Line 203: in the Gvirtzman et al. quote the comma should be deleted. Also, the capital T in The Zevulun Valley is unnecessary.

> **Reply:** Corrected, line 208. Thank you.

Line 255: station 123 is not visible in Figure S1.

> **Reply:** Thank you for pointing it out. We added station 123 to Figure S1.

Line 281 – it seems like there should be a better way to start a sentence than "Following, ". Perhaps: "We then examine the simulated…".

> **Reply:** Corrected, line 290. Thank you.

Page 14 – Fig. 6 – it seems that the SF scenario is missing from the figure altogether but referred to in the text. This is also true for figure 5. I also recommend that in the captions of figures 5 and 6 the abbreviations of the faults and source characteristics be explained again.

> **Reply:** Thank you for this comment. The SF (Shemona fault) scenario appears in figures 5 and 6 for M 7 scenarios. We did not model this scenario for M 6. Please refer to Table 3. We added additional description to these figures, as suggested, lines 266-268 (figure 5) and lines 314-316 (figure 6).

Line 299: the trade off sentence is a bit unclear (what trades off here). I suggest adding: "…. between the ground motion intensity in the Zevulun Valley (triangles) and the *ground motion intensity* in the Sedimentary wedge….".

> **Reply:** Modified. Thank you, see lines 306-313.

Line 300 – a comma is missing before "…in an asymmetric rupture…"

> **Reply:** Added, line 307. Thank you.

Figure 7 – the color bar of the mean peak ground velocity isn't very visible, I advise you to pick different colors, and make them vivid, say green – yellow – orange- red. Right now it is very difficult to observe the colors. Also, since you give just two locations as examples (station 129 and stations 127), it would be more convenient to identify these in the figure, rather than refer to a figure in the supplementary material.

> **Reply:** We modified the color bar and added stations 127 and 129 to the figure. Thank you.

Lines 369-375: this paragraph seems more suited to a summary section.

> **Reply:** Thank you for the suggestion. We believe that this paragraph contributes to the flow of the discussion and is an integral part of it. However, we renamed the section to "discussion and summary".

Line 399: the word is is missing: "noteworthy to mention **is** that…."

> **Reply:** Added, line 424. Thank you.

Line 404: when using the GMM acronym, the word model is redundant.

> **Reply:** Corrected, line 430. Thank you.

Line 406: define IM.

> **Reply:** Definition added, lines 170-171. Thank you.

Line 419: "four simulated ground motion**s** datasets" – the s of the plural for ground motion should be omitted.

> **Reply:** Corrected, line 448. Thank you.

[revised manuscript text omitted]

---

## Author Response (AR1)

Re: Authors Reply to Reviewers nhess-2021-280

Dear Prof. Frattini,

We would like to thank the reviewers of our paper, *"Ground motions variability in Israel from 3-D simulations of M 6 and M 7 earthquakes" (nhess-2021-280)"* for a constructive review, for the time spent and the positive attitude towards this work. The remarks and critique provided by the reviewers certainly improved the manuscript.

Attached below is our pointwise reply to the general and specific comments. All the changes and additions are marked in the revised version of the manuscript.

The co-authors approved the revision.

 Please address all correspondence to:

*Jonatan Glehman*

Department of Earth and Environmental Sciences
Ben Gurion University of the Negev
Beer-Sheva 8410501
Israel.

(glechman@post.bgu.ac.il)

Referee comments for manuscript nhess-2021-280 titled "Ground motions variability in Israel from 3-D simulations of M6 and M7 earthquakes", by Jonatan Glehman and Michael Tsesarsky

**General comments**

The manuscript describes a study of ground motion variability associated with simulations of 3D ground motions from M7 and M6 earthquakes, along major seismogenic sources in Israel. The authors use a 3-D model of the sub-surface that describes the main features of the spatial heterogeneity in Israel and simulate the ground motions at 129 measurement locations in the northern part of Israel. Then the authors derive a statistical attenuation model based on the simulations, and analyse the residuals, single station variability and significant durations.

This is of importance to low-moderate seismicity regions, and specifically to Israel. Complementing the instrumental catalogue with simulated strong ground motions is important, as well as understating and evaluating the variability associated with the simulated ground motions.

**Specific comments**

Why is there no subfigure for the minimization of the residuals to all 3 predictor variables in Figure S3 for magnitude 6?

> **Reply:** For M6, $Z_2$ has a minimal impact on the residual's standard deviation (<0.001), therefore we decided to not include it as an explanatory variable (please refer to table 4 and lines 415-418 in the revised manuscript); Hence, there was no subfigure for $Z_2$ in figure S3 for M 6.
> However, to clarify this point further, we followed your suggestion and added the quantitative impact of $Z_2$ on M 6 and the impact of Z2 on the minimization of the total residuals in Figure S3 (a,c and e). Thank you.

When comparing the simulations to CB14 – can the authors elaborate as to the predictor variable values used in CB14? Did they use 608 m/s for Vs30?

> **Reply:** Thank you for this comment. The CB-14 PGVs were calculated for a strike-slip fault, where we used the surface shear wave velocity as the Vs30 parameter input for CB-14 (this is due the minimum grid spacing of 76 meters). The Basin response term $Z_2$ in our model is analogue to $Z_{2.5}$ in CB-14. The Magnitude term was calculated according to the simulated magnitudes; Mw 6.21 and Mw 6.85. We added this information and an example of CB-14 GMM to Figure 5 to clarify this point further, please refer to lines 274 – 275 (Figure 5. Caption) and 345-347 in the revised manuscript.

It seems like the AM over-predicts GM at the sedimentary wedge rupture distances (Figure 9), can the authors comment on that? It would seem that most of the data at these distances is lower, and will result in a lower model that under-predicts.

> **Reply:** Thank you for this valuable comment. The residuals between the simulated and AM (mean PGVs in Figures 7 and 8 show that, except for stations 18 and 17 (located on the edge of the Sedimentary wedge), the AM under-predicts the ground motions at the Sedimentary wedge. Thus, the contribution to the AM's over-prediction comes from other sites located at these distances. As mentioned in lines: 356-359, the residuals were computed as a moving average over distance. Thus, they are independent of site-specific terms.

Lines 354 – 363: I really don't see this in the data. I don't see the triangles lower than the squares, and it is very difficult to tell them at all from the other symbols. I suggest thinking of a clearer way to present the data, as presently the figure does not support the statements in the text.

> **Reply:** Thank you for pointing it out. We modified figure 10 to support the text in the lines mentioned above (lines: 375-390 in the revised manuscript).

In lines 389 – 391, the authors state that the AM couldn't capture the full site effects of the Zevulun Valley and the sedimentary wedge, and further model refinements are required. Do the authors think that such effects can be incorporated successfully into a regional GMM?

> **Reply:** This is a challenging question. We do believe that such effects could be incorporated into a regional GMM. We are currently studying additional terms; unique to the Zevulun Valley and the Sedimentary wedge, such as $Z_{0.8}$ as well as distance-dependent and rupture velocity-dependent attenuation terms, for directivity and super-shear ruptures, among others. We added this information to the revised manuscript. Please refer to lines 475 - 478.

**Technical corrections**

The acronyms GMM and GMPE are used throughout the manuscript, please select one and be consistent.

> **Reply:** Thank you for this comment. We changed all acronyms to GMM.

The past and present tense are used interchangeably to describe what is / was done. Please select one and be consistent.

> **Reply:** Corrected. Thank you.

Line 70 – what is a magnitude limited GMPE? Unclear. Please explain.

> **Reply:** Explained. Please refer to lines 70-71 in the revised manuscript. Thank you.

Line 93: Israel should be Israeli.

> **Reply:** Corrected, line 99 in the revised manuscript. Thank you.

Line 93: ranges should be range.

> **Reply:** Corrected, line 99 in the revised manuscript. Thank you.

Figure 1 caption (lines 121, 123): I think it should say Israel seismic network **stations**.

> **Reply:** Corrected. Please refer to lines 127 and 128 in the revised manuscript. Thank you.

Line 135: "the Israeli coastal plain**,** " - the comma is missing, and the word "is" should be deleted.

> **Reply:** Corrected, line 141 in the revised manuscript. Thank you.

Line 143: delineate rather than delimit.

> **Reply:** Corrected, line 149 in the revised manuscript. Thank you.

Lines 159 – 166: the word rupture is misspelled many times.

> **Reply:** Corrected. Please refer to lines: 165-172 in the revised manuscript. Thank you.

Line 179: "ground motion**s** records"- the 's' should be deleted.

> **Reply:** Corrected, line 180 in the revised manuscript. Thank you.

Line 182: the sentence "we developed the regional velocity model of Shimony et al (2021) is unclear. If the model is Shimony et al.'s – then it wasn't developed by the authors. If it is a modification, or based on their results, then please explain what was done.

> **Reply:** Thank you for highlighting this point. The model of Shimony et al., was refined and expanded. We added this information to the text, lines 81-82 and 183-185 in the revised

Line 183: the word "following" here seems awkward. Please consider rephrasing.

> **Reply:** Corrected, line 185 in the revised manuscript. Thank you.

Line 185 – "statistical analysis **of** the synthetic database".

> **Reply:** Corrected, line 189 in the revised manuscript. Thank you.

Line 203: in the Gvirtzman et al. quote the comma should be deleted. Also, the capital T in The Zevulun Valley is unnecessary.

> **Reply:** Corrected, line 207 in the revised manuscript. Thank you.

Line 255: station 123 is not visible in Figure S1.

> **Reply:** Thank you for pointing it out. We added station 123 to Figure S1.

Line 281 – it seems like there should be a better way to start a sentence than "Following, ". Perhaps: "We then examine the simulated…".

> **Reply:** Corrected, line 295 in the revised manuscript. Thank you.

Page 14 – Fig. 6 – it seems that the SF scenario is missing from the figure altogether but referred to in the text. This is also true for figure 5. I also recommend that in the captions of figures 5 and 6 the abbreviations of the faults and source characteristics be explained again.

> **Reply:** Thank you for this comment. The SF (Shemona fault) scenario appears in figures 5 and 6 for M 7 scenarios. We did not model this scenario for M 6. Please refer to Table 3. We added additional description to these figures, as suggested, lines 270-272 (figure 5) and lines 320-321 (figure 6) in the revised manuscript.

Line 299: the trade off sentence is a bit unclear (what trades off here). I suggest adding: "…. between the ground motion intensity in the Zevulun Valley (triangles) and the *ground motion intensity* in the Sedimentary wedge….".

> **Reply:** Paragraph modified. Thank you, see lines 311-318 in the revised manuscript.

Line 300 – a comma is missing before "…in an asymmetric rupture…"

> **Reply:** Added, line 312 in the revised manuscript. Thank you.

Figure 7 – the color bar of the mean peak ground velocity isn't very visible, I advise you to pick different colors, and make them vivid, say green – yellow – orange- red. Right now it is very difficult to observe the colors. Also, since you give just two locations as examples (station 129 and stations 127), it would be more convenient to identify these in the figure, rather than refer to a figure in the supplementary material.

> **Reply:** We modified the color bar and added stations 127 and 129 to the figure. Thank you.

Lines 369-375: this paragraph seems more suited to a summary section.

> **Reply:** Thank you for the suggestion. We believe that this paragraph contributes to the flow of the discussion and is an integral part of it. However, we renamed the section to "discussion and summary".

Line 399: the word is is missing: "noteworthy to mention **is** that…."

> **Reply:** Added, line 434 in the revised manuscript. Thank you.

Line 404: when using the GMM acronym, the word model is redundant.

> **Reply:** Corrected, line 440 in the revised manuscript. Thank you.

Line 406: define IM.

> **Reply:** Definition added, line 282 in the revised manuscript. Thank you.

Line 419: "four simulated ground motion**s** datasets" – the s of the plural for ground motion should be omitted.

> **Reply:** Corrected, line 464 in the revised manuscript. Thank you.

**Referee Comments 2**

The authors use 3d simulation model to generate M6 and M7 seismic ground motion for Israel, considering local site effects, and source effects (directivity effects and supershear ruptures). Based on the generated data, they develop a local ground motion model (AM) M based solely on M6 and M7 and few rupture scenarios, and compare it with CB14 model.

The key objective of this work is not clear.

> **Reply:** We have tried to clearly state the objective, in the original manuscript (please refer to lines 167-172 and 415-416). We stress out that in this paper we only studied variability and didn't developed a GMM for Israel. This clearly reflects in the title of the paper. We accept this remark and emphasize the objective in the revised manuscript. Please refer to lines: 78-85 and 402-405 in the revised manuscript. Thank you.

The authors conclude that it is important to develop local GMM for Israel considering local sources, path and site effects.

> **Reply:** it is indeed one of our conclusions.

The authors do not explicitly show the simulation model used, nor the parameters or assumptions.

> **Reply:** We provide the essential model input in the original manuscript. Please refer to Figure 4 and Tables 1 (density, velocity, quality factors), 2 (grid and source time function) and 3 (simulated scenarios), in the revised manuscript. Also please refer to section 3.1 and lines 265-266 (section 4.1) in the revised manuscript for the assumptions. We believe the data provided sufficiently describes the numerical models and scenarios.

The authors do not validate the results of their simulations.

> **Reply:** Thank you for this comment. Strong motion instrumental record for Israel is not available (see the description in section 2.1), thus we cannot directly validate our model. The work of Shani-Kadmiel et al., (2016) who modeled the Jericho 1927 earthquake (the only M 6 earthquake in the catalogue) used similar numerical platform, similar source model (DSM) and a basic velocity model. The results of this work show good agreement between the reported and the calculated intensities. We consider this work as partial validation of our approach and model. We added this information to lines: 215-217 in the revised manuscript.

The authors claim that the results are model dependent (L414).

> **Reply:** Indeed. We state that to emphasize that the purpose of this study was not a development of an inclusive ground motion model but to study ground motion variability.

The authors do not compare their work with other papers in the region, or hybrid models (eg. please refer to Fayjaloun et al., 2021: Hybrid Simulation of Near-Fault Ground Motion for a Potential Mw 7 Earthquake in Lebanon).

**Reply:** Thank you for this comment. We would like to point out that the work of Fayjaloun et al., 2021 was published in October 2021, whereas our manuscript was submitted for review on 29.9.21. The comparison of simulated ground motions from different geological settings and different modeling assumptions (1-D to 3-D) may be limited. For example, several works showed that structural and material heterogeneity of the crust in Israel results in regional ground motions variability (Shimony et al., 2021; Volk et al., 2017). These can only be captured by 3-D modeling. We acknowledge the work of Fayjaloun et al., 2021 in the revised manuscript (lines: 454-458). As there are no other works of 3D simulated M 7 ground motions in Israel, we cannot compare our simulations with other databases. However, we compare our results with global GMPE's as they are based on the ergodic assumption and account for some site effects.

The authors do not clarify their choice of the attenuation functional model (please refer to http://www.gmpe.org.uk/gmpereport2014.pdf).

**Reply:** Thank you for this comment. The functional form of our attenuation model (AM) is based on the CB-14 function model (Campbell and Bozorgnia 2014). We also modified the $Z_2$ and $V_{s,ref}$ terms accordingly. Following your suggestion, we added this information to the revised manuscript, lines 279-288.

The authors validate the estimation of AM and of CB14 to the simulated GM: they find out that AM works better (which is obviously coming from the regression analysis using the same database) and conclude that CB14 ('imported GMM') deviates from the simulated GM.

**Reply:** Thank you for this comment. You are right. The comparison with AM is essential to show that it still deviates from the simulations even though it is based on it, and future refinements are needed. The CB-14 is indeed performing differently than our simulations. We do not compare the CB-14, and the AM as the AM was constructed not as a ground motion model but to examine the variability of the ground motions (lines 459-461 in the revised manuscript).

The authors do not justify the choice of CB14 model to compare their work in this region, considering that CB14 do not take directivity into account.

**Reply:** We choose CB-14 as it is a development of the CB-08 used in the Israel building code (413) and is planned to supersede it. We added this information to lines: 344-345 in the revised manuscript. Also, CB-14 is one of the NGA-West 2 based empirical ground-motion models widely used globally. To our best knowledge, none of the empirical models account for directivity. Thus, it is one of the reasons to develop a region-specific ground motion model accounting for such effects.

Please show: Gilboa and Carmel faults in figure 1.

**Reply:** Thank you for the suggestion. Gilboa and Carmel faults are parts of the Carmel Fault System (CFZ, please refer to lines: 101-102 in the revised manuscript) presented in Figure 1.

L97: the DSF magnitude potential of up to Mw7.5 (Hamiel et al., 2009).

**Reply:** Correct. We added the relevant value in line 103 in the revised manuscript. Thank you.

L 99- 101: the information are not fully coherent with Nof. et al 2021

(TRUAA—Earthquake Early Warning System for Israel: Implementation and Current Status).

> **Reply:** Thank you for pointing it out. We modified our manuscript accordingly. The updated information can be found in lines: 106-107 and 110-111 in the revised manuscript.

Better use the official name of the TRUAA project (instead of Tru'a)

> **Reply:** Thank you for pointing it out. Modified, lines: 110 and 128.

L 129: I would recommend the author to dedicate a few lines to better describe the spatial heterogeneity of the Earth structure.

> **Reply:** Thank you for this comment. please refer to section 2.2 "Spatial heterogeneity of Israel", in the revised manuscript.

L135: please add reference

> **Reply:** Added to line 148 of the revised manuscript. Thank you.

L189: I would recommend a few lines to describe the software, the (dynamic ?) simulations, how does it consider the source, propagation and site effects, the assumptions made, the choice of the nucleation point

> **Reply:** Thank you for this comment. The necessary information was provided in section 3.1 "Numerical model" and tables: 1,2 and 3.

L 234-236: the authors choose rupture speed to be equal to 0.9Vs and 1Vs for subshear and supershear scenarios respectively. please justify this, knowing that the rupture speed should be lesser than 0.85Vs or larger than 1.2Vs.

> **Reply:** Rupture speed values usually vary between 0.6-0.9 Vs. (see for example Heaton, 1990). We chose 0.9Vs, as this value is widely used in the literature, see for example (not an inclusive list) works of: Kaneko and Shearer (2014); Bizzarri and Spudich, (2008); Lin et al., (2020); Liu et al., (2014); Weng and Ampuero (2020). As per the supershear speed, the rupture nucleates within the hard rock with a subshear speed of 1800 m/s. It evolves into supershear rupture when it ruptures the sediments with shear wave velocity of <900 m/s (greater than the Eshelby speed 1.41Vs, as predicted for supershear ruptures). Please refer to lines: 241-243 in the revised manuscript.

L252: why 129 GM simulations seem sufficient?

> **Reply:** We have produced 129 synthetic records for each simulation (five for each magnitude). Total of 633 and 645 ground motion records for M 7 and M 6 scenarios. First, we deployed a uniform grid with 10 km spacing (total of 124 synthetic stations). Following we added five more stations in areas of interest (such as Zevulun Valley, Kiryat Shemona, among others). This deployment provides sufficient spatial coverage to account for ground motion variations. The ground motion model will be further constrained with more ground motion records. We added this information to lines 187-189 in the revised manuscript.

L 352: Can the authors explain why the duration of the simulated GM is not function of the distance?

**Reply:** Thank you for this comment. We added this information to lines 377-379 and 448-450 in the revised manuscript.

L 399: please explicity show how you notice this conclusion from the AM. do you notice the same conclusion with your simulated ground motion ?

**Reply:** Thank you for this comment. We modified Figure 6 to represent our results better, and to properly differentiate between residuals associated with source effects from those associated with path and site effects. The AM adopts a typical form of ground motion model. Figures 6c,d show that the within-events residuals ($\delta W$) are path-dependent for the directivity and supershear scenarios (lines: 308-310 in the revised manuscript). Whereas Figures 6e,f clearly show that the between events residuals ($\delta B$) related to source effects are not zero for the supershear scenario (however, they are equal to zero for the directivity scenario, lines 317-318 in the revised manuscript). This means that additional path terms should be incorporated in the AM for the directivity and supershear ruptures, and additional source terms for supershear ruptures. Please refer to lines: 409-413 and 434-439 in the revised manuscript.

L 415-416: you can not study the variability of the ground motion with a model that is not validated.

**Reply:** Thank you for this comment. As we explained above the main challenge in Israel is the low seismicity rate which results in a M >6 gap in the instrumental catalogue. Using low magnitude events would not provide the necessary validation as the details of finite fault kinematics of M > 6 earthquakes are not covered by small (nearly point) source models. Furthermore, the spatial coverage of the ISN (at least till 2020) is sparse and doesn't cover areas of interest. This being the case, validation *sensu-stricto*, is not within reach. We do report the Shani-Kadmiel (2016) work as a benchmark. We added this information to lines: 215-217 in the revised manuscript. Numerical models are an essential tool to fill in the sparse data and knowledge gaps. Here, we further develop the regional velocity model, both near the faults and in the regions of interest to provide a more refined model. This model clearly shows ground motions variability, compared to reference models. We report the results with the necessary caution and do not claim that these results should (or could be used) as is. We further stress out that our purpose was not to develop a GMM but rather study the sources for ground motion variability (lines 460-461 in the revised manuscript). We do plan to develop and validate a hybrid (instrumental – numerical) GMM in the future (472-480 in the revised manuscript).

L434: this statement is not a result of your work and thus should not be described in the summary section.

**Reply:** we accepted the suggestion and rewrote the paragraph. Please refer to lines: 481-488 in the revised manuscript.

figure 1:

• better resolution ?

**Reply:** modified. Thank you.

• b. please define PA in the caption.

**Reply:** Added. Line 126 in the revised manuscript. Thank you.

• b. I would recommend the authors to change the description: 'the Israel seismic network in Israel: yellow.. and brown .. . the green circles show the population ..'

> **Reply:** Thank you for the suggestion. However, we believe that the current formulation contributes better to the caption's flow.

figure 2:

plot Y in logarithmic scale ?

> **Reply:** Thank you for the suggestion. However, there is no need for a logarithmic scale as Y-axis values are of one order of magnitude.

figure 3:

▪ show the location of the vs profile on plot (a)

> **Reply:** Locations added. Thank you.

▪ what is the reference of this plot what do the yellow and purple colors represent ?

> **Reply:** Thank you for this comment. The color description was added to lines 174-175 in the revised manuscript. For references to the plot, please refer to section 2.2, "Spatial heterogeneity of Israel."

figure 5, 6, 10: use the same color legend for M6 and M7. figure 10: remove 'comparison' from the caption description.

> **Reply:** Color legend modified. Different colors assigned for CFZ (M 6) and SF (M 7) scenarios. We rewrote the caption's description. Please refer to line 391 in the revised manuscript. Thank you.